**PLOS** NEGLECTED TROPICAL DISEASES

# The potential impact of human visceral leishmaniasis vaccines on population incidence

**Epke A. Le Rutte**[1,2,3]*, **Luc E. Coffeng**[1], **Stefano Malvolti**[4], **Paul M. Kaye**[5], **Sake J. de Vlas**[1]

**1** Department of Public Health, Erasmus MC, University Medical Center Rotterdam, Rotterdam, The Netherlands, **2** Department of Epidemiology and Public Health, Swiss Tropical and Public Health Institute, Basel, Switzerland, **3** University of Basel, Basel, Switzerland, **4** MM Global Health Consulting, Zurich, Switzerland, **5** York Biomedical Research Institute, Hull York Medical School, University of York, York, United Kingdom

* e.lerutte@erasmusmc.nl

**Data Availability Statement:** All relevant data are within the manuscript.

**Funding:** EALR, LEC, and SJDV gratefully acknowledge funding of the NTD Modelling

## Abstract

Human visceral leishmaniasis (VL) vaccines are currently under development and there is a need to understand their potential impact on population wide VL incidence. We implement four characteristics from different human VL vaccine candidates into two published VL transmission model variants to estimate the potential impact of these vaccine characteristics on population-wide anthroponotic VL incidence on the Indian subcontinent (ISC). The vaccines that are simulated in this study 1) reduce the infectiousness of infected individuals towards sand flies, 2) reduce risk of developing symptoms after infection, 3) reduce the risk of developing post-kala-azar dermal leishmaniasis (PKDL), or 4) lead to the development of transient immunity. We also compare and combine a vaccine strategy with current interventions to identify their potential role in elimination of VL as a public health problem. We show that the first two simulated vaccine characteristics can greatly reduce VL incidence. For these vaccines, an approximate 60% vaccine efficacy would lead to achieving the ISC elimination target (<1 VL case per 10,000 population per year) within 10 years' time in a moderately endemic setting when vaccinating 100% of the population. Vaccinating VL cases to prevent the development of PKDL is a promising tool to sustain the low incidence elimination target after regular interventions are halted. Vaccines triggering the development of transient immunity protecting against infection lead to the biggest reduction in VL incidence, but booster doses are required to achieve perduring impact. Even though vaccines are not yet available for implementation, their development should be pursued as their potential impact on transmission can be substantial, both in decreasing incidence at the population level as well as in sustaining the ISC elimination target when other interventions are halted.

Consortium by the Bill and Melinda Gates Foundation (OPP1184344). LEC further acknowledges funding from the Dutch Research Council (NWO, grant 016.Veni.178.023). PMK and SM are supported by The Welcome Trust (grant numbers WT108518 and WT1063203). The funders had no role in study design, data collection and analysis, decision to publish, or preparation of the manuscript.

**Competing interests:** I have read the journal's policy and the authors of this manuscript have the following competing interests: SM / MMGH received funding from the University of York to support the definition of a clinical development strategy and related business case for the vaccine candidate ChAd63-KH. PMK is funded by the Wellcome Trust (WT108518) and EDCTP (RIA2016V-1640) to develop a vaccine against leishmaniasis and is co-inventor of a patent covering a synthetic vaccine gene sequence (PCT/GB2010/000815).

## Author summary

Vaccines for human visceral leishmaniasis (VL) are currently under development. In this study, we simulate VL transmission dynamics using mathematical models to explore the potential impact of vaccines on population-wide incidence. We show that some vaccines have high potential to reduce VL incidence, namely those that reduce the infectiousness of infected individuals to sand flies and those that reduce the chance of developing symptoms once infected. The effect of vaccines that lead to protection from infection is potentially the greatest, but depending on the duration of immunity, individuals would require booster doses to guarantee lifelong impact. Vaccines that prevent the development of post-kala-azar dermal leishmaniasis are a promising tool to sustain low VL incidence and prevent recrudescence of infection when regular interventions are halted. Our results strongly support the continued development of VL vaccines, as their potential impact on population incidence can be substantial.

## Introduction

Visceral leishmaniasis (VL), also known as kala-azar, is a vector-borne neglected tropical disease. Infection occurs after successful transmission of the *Leishmania* protozoa through the bite of an infected female sand fly [1]. Most infected humans remain asymptomatic, and only a small proportion of about 1–10% develop clinical symptoms, resulting in death when left untreated [2,3]. Between 5% and 20% of treated VL cases develop a long-lasting skin condition known as post-kala-azar dermal leishmaniasis (PKDL). Recent studies have identified that individuals with PKDL are equally infectious towards sand flies as VL cases, making them an important reservoir of infection [4,5]. However, the contribution of asymptomatic individuals to transmission has not yet been defined [4,6]. After infection, a period of immunity follows, of which the duration remains debated.

Currently around 33,000–66,000 individuals develop symptomatic VL each year, mainly on the Indian subcontinent (ISC), Eastern Africa, the Mediterranean region, and Brazil, affecting the poorest of the poor [7,8]. The World Health Organization (WHO) and affected countries target for 'elimination of VL as a public health problem by 2020' on the ISC, where VL is considered to be solely anthroponotic. This target is defined as maintaining less than 1 VL case per 10,000 individuals per year at district level in Nepal, at subdistrict/block level in India, and at upazila level in Bangladesh [9]. In the rest of the world (e.g Africa, Europe, Brazil), where VL can also be zoonotic with the main reservoir of infection in dogs, the target is 100% detection and treatment of symptomatic cases [10]. Current strategies consist of diagnosis and treatment of VL cases, and vector control.

Vaccines already play an important role in the control of canine leishmaniasis, at the individual level they reduce the development of symptoms, reduce the parasite load in the blood, and reduce the risk of death [11–13]. These vaccines have also proven to be effective at the population level by reducing *Leishmania* transmission, resulting in lower incidence in both dogs and humans [14,15]. The development of human VL vaccines has been on-going for decades and there are different vaccine candidates currently in trial, but none are yet available for implementation [16,17]. The promising results from experimental human VL vaccine trials, and by the practice of "leishmanization", in which a healthy individual is artificially exposed to tissue scrapings derived from a cutaneous leishmaniasis patient, leading to disease prevention [6,16,18–20], provide strong evidence for the scientific feasibility of an effective vaccine against human VL. Should an effective vaccine become available, it has been estimated

to be cost-effective when used at large scale and in addition to ongoing diagnosis and treatment, without even accounting for its impact on transmission [19].

Mathematical transmission models are useful tools to gain insight into the effect of current and future interventions on VL incidence and the underlying transmission dynamics. Previous modelling studies that focused on VL transmission on the ISC presented two model variants; one in which only VL and PKDL cases contribute to transmission, and another in which also asymptomatic individuals contribute to transmission (~1% relative to VL cases). The models estimated that in most situations on the ISC, the target is likely to be met with current strategies but in high endemic settings and at a lower geographical scale, additional efforts are required. They also highlighted the risk of recrudescence of infection after achieving the low incidence target, when halting interventions. This is mainly due to individuals with PKDL and/or asymptomatic infection. Therefore, the studies emphasized the need for further research on the potential impact of preventive VL and PKDL strategies as a tool in reaching and sustaining VL elimination as a public health problem on the ISC [5,21,22]. Other studies stressed that 100% detection and treatment of cases in the rest of the world remains challenging and that prevention could be much more effective than case detection and treatment [23].

In this study, we implement multiple characteristics of potential human VL vaccines using the two variants of a deterministic VL transmission model [21] to estimate the potential impact of these vaccine characteristics on VL incidence and transmission dynamics during and after the achievement of the current elimination target. The vaccines that are simulated in this study 1) reduce the infectiousness of infected individuals towards the sand fly, 2) reduce the risk of developing symptoms after infection, 3) reduce the risk of development of PKDL, or 4) lead to the development of transient immunity to infection [24–26]. We also compare and combine vaccine characteristics with current interventions to identify which vaccines could be most impactful in fighting this neglected tropical disease.

## Methods

### Overview of VL vaccine candidates and characteristics

Currently there are various VL vaccine candidates under study [27]: LEISH-F3+GLA-SE [28,29], and ChAd63-KH (ISRCTN07766359) [30] are currently in clinical development; Ad5-A2/rA2 Prime / Boost [31], genetically modified live attenuated whole parasites [25,26,32], and a LmCen$^{-/-}$ vaccine [33] are being developed for the clinic [34].

These vaccines have different physical and immunological properties, and could be used in either prophylactic or therapeutic settings, but their impact following infected sand fly bite in humans has yet to be evaluated. Table 1 summarizes different potential vaccine outcome measures (herein called characteristics) that were selected for simulation in this study. Vaccine characteristic 1 is separated into 1a) asymptomatic individuals and 1b) all infected individuals, because it is suggested that only individuals with asymptomatic infection may be affected by the vaccine and that once an individual develops symptoms there are no differences in

**Table 1. Human VL vaccine characteristics.**

| Number | Vaccine characteristic |
| --- | --- |
| 1a | Reduced infectivity of asymptomatic individuals |
| 1b | Reduced infectivity of all infected individuals |
| 2 | Reduced risk of developing symptoms |
| 3 | Reduced risk of developing PKDL |
| 4 | Development of transient immunity protecting against infection |

infectiveness (1a). However, since this is not yet well established, we also include the option where all infected individuals become less infective, as a result of the vaccine (1b).

## Transmission models and simulation of existing interventions

Fig 1 illustrates the basic structure of the VL transmission model, which is a deterministic age-structured model. There are two model variants, that only differ based on assumptions about where the main reservoir of infection lies; namely, solely in symptomatic individuals (VL and PKDL), or mainly in asymptomatic individuals [21,35,36]. The models were parameterized with age-structured data on approximately 21,000 individuals included in the KalaNet bednet trial in India and Nepal [37] and have undergone geographical cross-validation against data on >5000 VL cases from 8 endemic districts in Bihar collected by CARE India [38] (see [36]

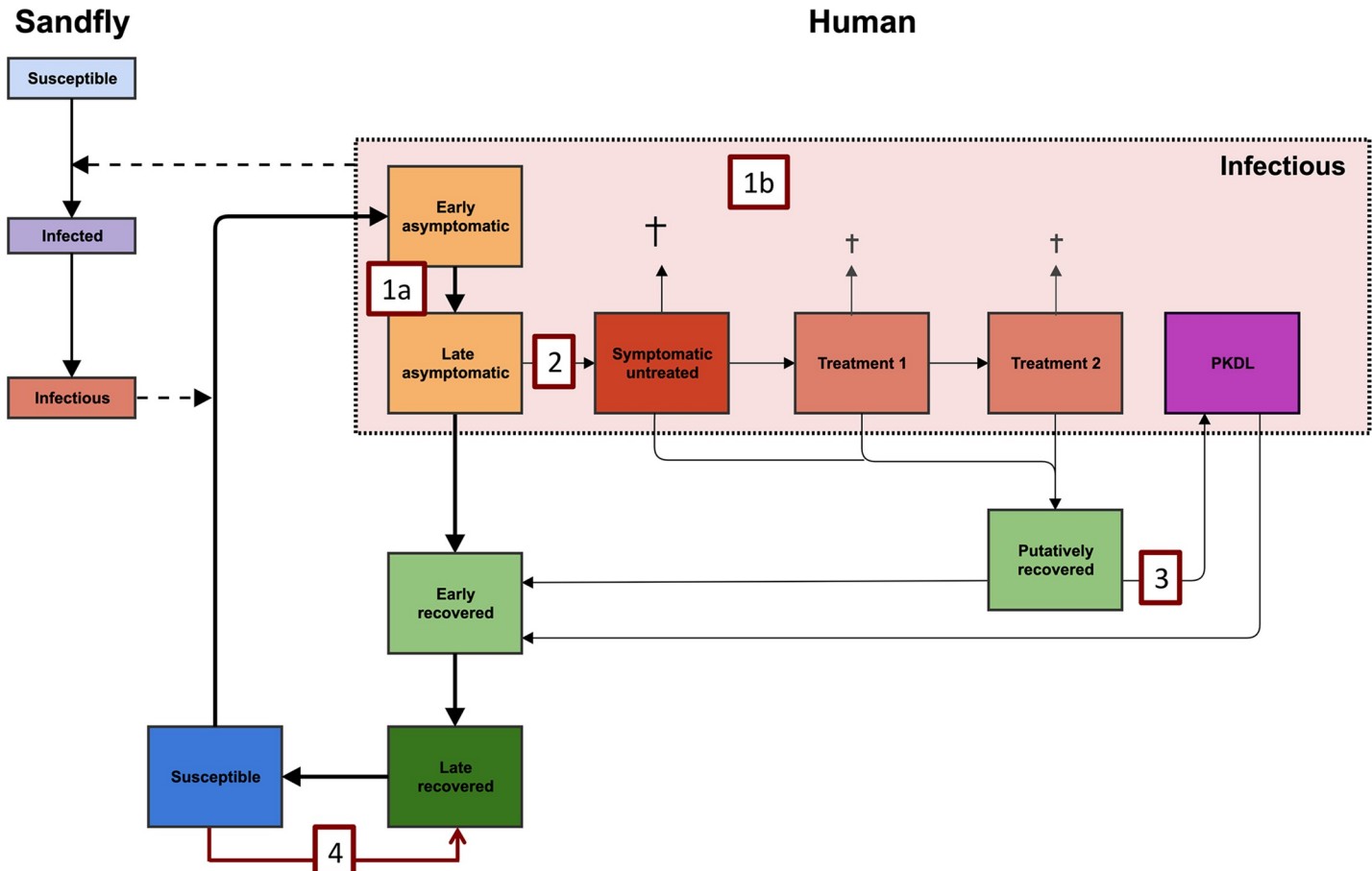

**Fig 1. Schematic presentation of the model variant in which asymptomatic individuals contribute to transmission, with numbers related to different types of vaccine characteristics that are implemented in the models.** In the alternative model variant, asymptomatic individuals are assumed not to be infectious towards to sand fly, with infection pressure only coming from symptomatic individuals with VL (with and without treatment) and PKDL. Once a susceptible individual is infected by an infectious sand fly, they become early asymptomatic for an average duration of about 200 days, which is followed by the late asymptomatic stage (average duration of 69 days). The average infectivity of both asymptomatic stages together is 0 in the model in which they do not contribute and ~1.5% relative to VL in the model in which they contribute to transmission. 1.4% of late asymptomatic individuals develops VL, and without active case detection, the duration between onset of symptoms and start of treatment lasts on average 40 days, followed by 1-day treatment 1 and potentially 28-day treatment 2 or death if left untreated. The average duration of the putatively recovered stage is 21 months and 5% of these individuals develop PKDL which lasts 5 years on average. The infectivity of PKDL is 90%, relative to VL. The rest recovers to the early recovered stage (average duration of 74 days), followed by the late recovered stage (average duration of 2 years), which can be interpreted as the duration of immunity. The numbers in the red boxes relate to the numbers in the first column of Table 1 and represent the following vaccine characteristics; 1a) early and late asymptomatic individuals become half as infectious, 1b) all infection states become half as infectious, 2) vaccinated individuals are 50% less likely to develop symptoms, 3) vaccinated individuals are 50% less likely to develop PKDL, and 4) vaccinated individuals develop transient immunity against infection.

for full model code and descriptions, and sensitivity analyses). Recent outcomes from xenodiagnosis studies have been incorporated, indicating that those with PKDL are on average nearly as infectious as those with VL (0.9:1.0) [4,5].

Interventions of which the effects have previously been modelled are vector control through indoor-residual spraying of insecticide (IRS) and active case detection (ACD). The guidelines, as developed by WHO, recommend a 5-year attack phase (intense IRS and ACD) followed by 5 years of consolidation phase (IRS and intense ACD). In our models, IRS leads to a decrease in sand fly density and ACD shortens the duration of the symptomatic untreated stage.

## Implementation and simulation of four vaccine characteristics

Vaccine characteristic 1 is simulated by a reduction in infectiousness of infectious states towards the sand fly. For vaccine characteristics 2 and 3, the respective flow towards clinical VL and PKDL is reduced. With vaccine characteristic 4, we selected 100% development of transient immunity after having received the vaccine and experimented with vaccinating 100% and 50% of the population. The duration of immunity after vaccination is assumed to be to 2 years, which is similar to the assumed duration of immunity after natural infection in our model of which sensitivity analyses are presented in previous work [36].

For the simulations of vaccine characteristics, we assume that they apply to everyone involved, i.e. all ages and sexes. No specific target populations are simulated, besides for vaccine characteristic 3, which is only administered to those that have developed VL. For vaccine characteristics 1–3, we assume an arbitrary 50% reduction of the infectiousness as well as a 50% reduction of the proportions of individuals that develop VL and PKDL, all in combination with a 100% vaccination coverage. We also calculate the percentage of vaccine effectiveness required to achieve the VL elimination target incidence of 1/10,000/year within 10 years of starting the intervention, which could aid in defining a vaccine target product profile (TPP). We assume that the vaccine characteristics are in place constantly from the start of the intervention, except for vaccine characteristic 4, where we experiment with simulating a single vaccination round and repeated yearly vaccination rounds. For all four vaccine characteristics, we separately simulate and compare their impact on VL incidence over time, even though it is likely that one vaccine will possess multiple characteristics. The cumulative effects of some vaccine characteristics are simulated indirectly, as reducing the development of VL will lead to a decrease in the overall development of PKDL. Previous work has shown that when current existing interventions have led to the 1/10,000/year target, there are many susceptible individuals and the infection pressure comes mainly from PKDL cases (when assuming the infection pressure originates from symptomatic individuals only) [21]. To address this, we also combine vaccine characteristic 3, vaccination of VL cases to prevent the development of PKDL, with the current interventions recommended by WHO.

## Results

The impact of each of the four vaccine characteristics on VL incidence is illustrated in Fig 2. A vaccine that reduces infectivity of asymptomatic individuals by 50% (1a) leads to achievement of the target of less than 1 VL case per 10,000 population per year in about 11 years. When all infected individuals have a reduced infectiousness of 50% (1b), the decline is steeper, achieving elimination in around 4 years if asymptomatics are the main reservoir of infection and 11 years when infection is only coming from those with VL and PKDL. Halving the chance of developing symptoms (2) also has a considerable impact on transmission, especially if only symptomatic individuals are infective after which elimination takes about 10 years. However, if most infection pressure arises from asymptomatic individuals, the impact of halving the

### Model predictions

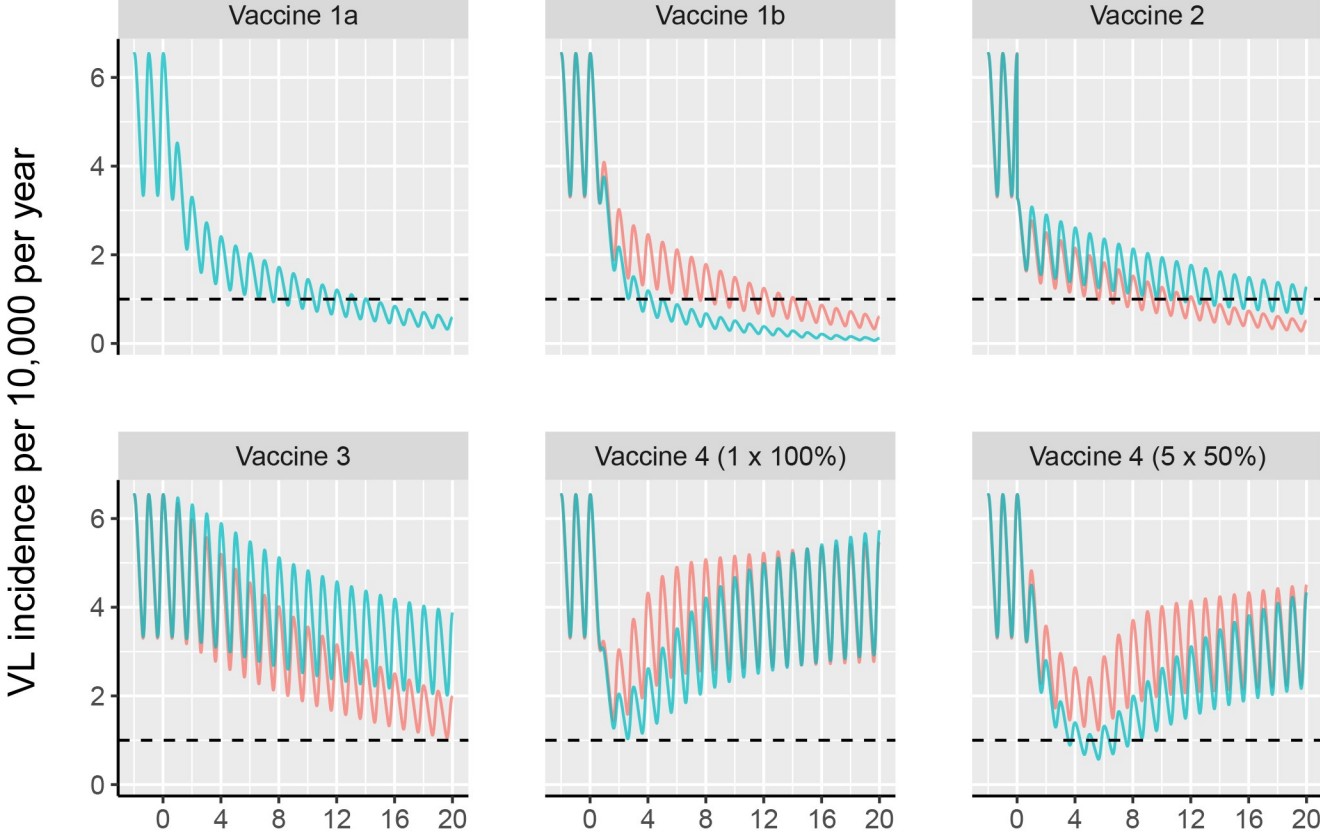

**Fig 2. The impact of different vaccine characteristics on VL incidence using model variants with and without asymptomatics contributing to transmission in a setting with a pre-control endemicity of 5/10,000/year.** Vaccine characteristics are in place continuously from year 0 onwards, unless for vaccine characteristic 4, which is administered once (1 x 100%), or yearly for five years in a row (5 x 50%). The different vaccine characteristics that are also explained in Table 1 and illustrated in Fig 1, are; 1a) early and late asymptomatic individuals become half as infectious, 1b) all infection states become half as infectious, 2) vaccinated individuals are 50% less likely to develop symptoms, 3) vaccinated individuals are 50% less likely to develop PKDL, and 4) vaccinated individuals become immediately immune. The black dashed line represents the WHO elimination target of 1/10,000/year. The oscillations in VL incidence are a result of seasonality in the sand fly density.

development of symptoms will lead to achieving the elimination target only after about 19 years, when used as a stand-alone tool. A 50% reduction in the development of PKDL (3), after which not 5% (default) but only 2.5% of past VL cases develop PKDL, has the smallest impact on transmission. As expected with this characteristic, the relatively larger impact is seen when only those with VL and PKDL contribute to transmission, and thus when PKDL plays a more prominent role in the transmission dynamics. Of all vaccine characteristics, the development of immunity that protects against infection (as seen in late recovered cases) of the population causes the most rapid decrease in incidence (4), since the pool of susceptible individuals is completely removed at once (with the assumption of 100% coverage as used in the model). We

**Table 2. Minimum required effect of the vaccine characteristics to reach a VL elimination target incidence of 1/10,000/year within 10 years' time after starting the intervention, when vaccinating 100% of the population in a setting with a 5/10,000/year pre-control incidence.**

| Vaccine characteristic | Model variant | |
|---|---|---|
| | Only VL and PKDL contribute to transmission | Asymptomatics are main contributors to transmission |
| 1a) required reduction in infectivity of asymptomatic individuals | N/A | 35% |
| 1b) required reduction in infectivity of all infected individuals | 60% | 37% |
| 2) required reduction in the development of symptoms | 56% | 68% |
| 3) required amount of time to reach the elimination target when preventing the development of PKDL completely | 11 years | >20 years |
| 4) required minimum number of rounds when vaccinating 50% of the susceptible individuals yearly with 100% vaccine efficacy | 14 rounds | 5 rounds |

additionally explored the effect of vaccinating half the population and repeating this yearly for 5 years in a row (5 x 50%), showing that regular vaccinations are required to sustain the impact and move towards the low incidence elimination target.

The minimum vaccine effect required for each vaccine characteristic to achieve the VL elimination target incidence of 1/10,000/year within 10 years of starting the intervention is presented in Table 2. The vaccine characteristics that impact the development of VL and PKDL (2 and 3) obviously have a bigger impact in the model in which only VL and PKDL contribute to transmission.

Vaccine characteristic 3, after which vaccinated individuals are less likely to develop PKDL, displayed the least impact when used as a stand-alone tool. Fig 3 shows the impact on VL incidence of a decrease in the development of PKDL of 50% and 100%, combined with the current interventions for a setting with a pre-control endemicity level of 5/10,000/year. The red line represents the default scenario in which the current interventions (active case detection and vector control) are in place during the WHO attack phase (year 0–5) and the WHO consolidation phase (year 5–10), without the presence of a vaccine. Further details on the impact of current interventions on VL incidence on the ISC as predicted by these models can be found in Le Rutte *et al.*, 2018 [21]. After halting all interventions at year 10, the situation will slowly return to the pre-control equilibrium of 5/10,000/year, because of the remaining VL incidence in year 10 in all scenarios. In the two scenarios with the PKDL vaccine (green and blue lines) a new, much lower, equilibrium will be reached after regular interventions are halted. For the vaccine with a 50% efficacy (50% decrease in PKDL development of vaccinated VL cases) the target of 1/10,000/year will be reached as simulated by the model in which only VL and PKDL contribute to transmission. When assuming an effect of 100% protection from developing PKDL, this model suggests that using only vaccine 3 could keep the incidence below 1/10,000/year, after all regular interventions have brought incidence down and are halted. However, in settings with a higher pre-control endemicity of 10/10,000/year, only the vaccine with 100% protection against development of PKDL will lead to the elimination target of VL after 15–20 years depending on the start year of the PKDL vaccine.

## Discussion

In this study, we present for the first time the potential impact of VL vaccines on transmission dynamics and population incidence on the Indian subcontinent (ISC). This impact looks very promising. We found that all simulated vaccine characteristics show potential in reducing population VL incidence, particularly those that reduce the infected individual's infectiousness or reduce the chance of developing symptoms once infected. For these vaccines, an approximate

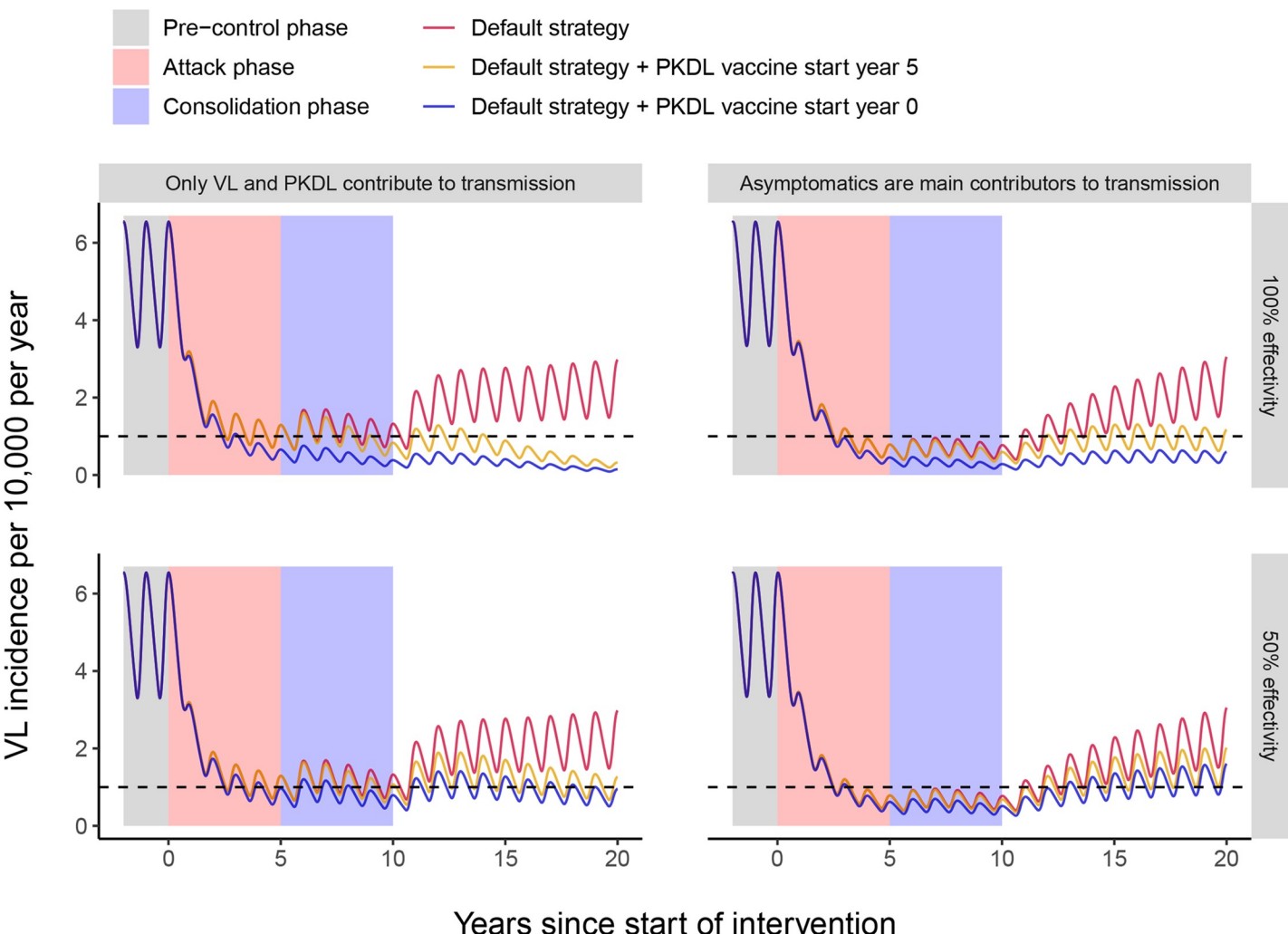

**Fig 3. Strategies of combining vaccine effect 3 with the WHO attack and consolidation phase for a setting with a pre-control endemicity level of 5/10,000/year. Top panels: vaccine effect with 100% protection against the development of PKDL, bottom panels: vaccine effect with 50% protection against the development of PKDL.** The default strategy is visualized with the red line (top and bottom row identical), in which 5 years of attack phase are followed by 5 years of consolidation phase, after which interventions are halted in year 10. For the green line, the PKDL vaccine is introduced during the consolidation phase (year 5), which continues after the consolidation phase has ended at year 10. For the blue line, the PKDL vaccine is already introduced at the start of the attack phase (year 0), continues during the consolidation phase and is continued when regular interventions are halted in year 10. Left figures show the simulations for the model variant where solely symptomatic individuals contribute to transmission, whereas for the right figures asymptomatic individuals constitute the main reservoir of infection. The black dashed line represents the WHO VL incidence target of 1/10,000/year. The oscillations in VL incidence are a result of seasonality in the sand fly density.

60% vaccine efficacy would lead to achieving the ISC elimination target (<1 VL case per 10,000 population per year) within 10 years' time in a moderately endemic setting, assuming that the entire population is vaccinated and only VL and PKDL cases contribute to transmission. For the model variant in which asymptomatics are the main contributors to transmission, much lower vaccine efficacies of around 37% would be required when reducing the infectiousness; however, for the required reduction in the development of symptoms, a vaccine efficacy of nearly 70% was estimated. The vaccine that leads to immunity akin to that of late recovered cases shows the highest impact, but individuals would require regular booster vaccines to achieve and sustain the low incidence elimination target. Vaccinating VL cases to prevent the development of PKDL shows to be a promising tool to sustain the elimination target once reached, and prevent recrudescence of infection when regular interventions are halted. Those

findings are of great importance in providing a factual base to the ongoing effort aimed at establishing a TPP for a VL vaccine.

A limitation to our study is the fact that we simulated vaccine characteristics rather simplistically by instantaneously altering the transition rates and applying this simultaneously to all individuals in the population. Ideally, vaccinated individuals should move to different, additional, compartments in the model, where they experience a different history of infection. In such a model, vaccinated and unvaccinated individuals would be living beside each other, both influencing the transmission dynamics differently, although the outcomes would likely only differ quantitatively with ours. Another limitation of our study is that we only present the results for a setting with a pre-control endemicity of 5 VL cases per 10,000 population per year, which we considered representative for endemic situations where vaccines would be most useful. In settings with a lower pre-control endemicity the elimination target would be achieved earlier; in settings with a higher pre-control endemicity, the vaccine characteristics would require a higher efficacy to achieve the same effect on VL incidence in the same amount of time.

We further decided to simulate the vaccine characteristics separately, while in reality most vaccines are expected to possess multiple characteristics. For example lowering the parasite load will likely lead to both decreased infectiousness as well as reduced development of symptoms, as is also seen in canine VL vaccines [12,13]. However, by combining them it would be less clear to what extent different characteristics would drive the total impact of a vaccine. For the vaccine that causes vaccinated individuals to develop transient immunity against infection, it is important to note that the impact on VL incidence, as well as the required number of booster vaccines, highly depends on the duration of acquired immunity, which was assumed to be two years on average in our models similar to what we used in previous work [36]. The longer the duration of acquired immunity, the bigger the impact on VL incidence and the lower the frequency of required booster vaccines. We also assume that for all vaccine characteristics the efficacy is 50%. Even though this is a generalization and in reality it is likely different for each characteristic, this approach allows us to compare the impact of the different vaccine characteristics. In this study we simulate transmission between humans and sand flies, which is currently considered to reflect the transmission dynamics of VL on the ISC. However, would a considerable contribution to transmission come from an animal reservoir, vertical transmission as seen in dogs, and/or the presence of those with HIV-VL co-infection, the potential impact of vaccines could increase [13,39–41].

A typical aspect of the deterministic model that we use is that all durations of states are exponentially distributed, which often does not reflect the actual distributions of durations as found in nature. The slow recrudescence of infection between year 10 and 20 is another phenomenon of the deterministic model, where prevalences can never become completely zero, but in reality the disease will either die out or come back, and if it comes back, most likely it will progress somewhat faster. Around the elimination target when numbers of infected cases become very low, the role of chance increases and a stochastic transmission model would be required to analyse the risks of recrudescence or the chance of achieving (local) elimination of transmission.

We acknowledge that some of the assumptions chosen for the simulation are not fully reflective of the reality of implemented immunization programs. Firstly, our choice of 100% coverage certainly is an overestimation of what can be realistically achieved. For example, coverage for the 1st dose of measles-containing vaccines was on average 73% in the AFRO region, and the human papilloma virus vaccination had an average coverage of 88% when pooling regions and income levels [42,43]. However, this assumption allowed us to evaluate the maximum impact and to do such an evaluation independently from the constraints of delivery strategy. Also, having chosen another coverage level would not alter our outcomes when

comparing the impact of the different characteristics. With a lower coverage, the durations until elimination as a public health problem would be longer and the minimum required efficacies would be higher. Secondly, and in particular at the start of vaccination programs, a catch up campaign is usually implemented to quickly reduce the susceptible population, focusing on the population that is at highest risk (i.e. for leishmaniasis, children and young adults or migrant workers [44]). Such programmatic design considerations are not considered in the current model and will need to be investigated with more complex individual-based transmission models. Lastly, 5-year protection is most likely going to be the minimum requirement to allow for a widespread roll-out in routine immunization. Shorter durations requiring a very frequent administration of booster doses might prove programmatically and financially unsustainable. Nevertheless, from an impact assessment standpoint the results generated with the more conservative assumptions of the current model have clear significance for understanding the relative importance of different vaccine characteristics.

Vaccines have proven to be vital tools in the control and prevention of diseases [45,46]. This study reveals that a VL vaccine strategy could also prove an important tool in the fight against this neglected tropical disease. We focussed on the anthroponotic transmission dynamics of VL on the Indian subcontinent, but also in the rest of the world VL vaccines are likely to surpass their impact at the patient level by reducing the infection pressure, positively impacting the estimated 6 million people at risk of VL globally [47].

In conclusion, even though VL vaccines are not yet available for implementation, our results strongly support their continued development, given the potentially substantive impact on transmission, decreasing incidence at the population level, and sustaining the low incidence elimination target on the ISC when other interventions are relaxed. More details of the impact of different vaccines characteristics on the history of infection are awaited to further our understanding and modelling of the impact of VL vaccines on VL transmission dynamics and disease incidence.

## Acknowledgments

We are very grateful for having had the opportunity to learn from the insights and tremendous expertise of Professor Farrokh Modabber while discussing this work. We would also like to thank the organizers and attendees of the VL vaccine expert meeting in Rockville, USA, in September 2015 at the National Institute of Allergy and Infectious Diseases, where the idea for this study sparked.

## Author Contributions

**Conceptualization:** Epke A. Le Rutte, Sake J. de Vlas.

**Formal analysis:** Epke A. Le Rutte.

**Investigation:** Epke A. Le Rutte.

**Methodology:** Epke A. Le Rutte, Luc E. Coffeng.

**Supervision:** Sake J. de Vlas.

**Validation:** Stefano Malvolti, Paul M. Kaye.

**Visualization:** Epke A. Le Rutte.

**Writing – original draft:** Epke A. Le Rutte.

**Writing – review & editing:** Luc E. Coffeng, Stefano Malvolti, Paul M. Kaye, Sake J. de Vlas.

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
