## [Decision Letter · Decision Letter 0]

5 May 2020

Dear Dr Le Rutte,

Thank you very much for submitting your manuscript "The potential impact of human visceral leishmaniasis vaccines on population incidence" for consideration at PLOS Neglected Tropical Diseases. As with all papers reviewed by the journal, your manuscript was reviewed by members of the editorial board and by several independent reviewers. The reviewers appreciated the attention to an important topic. Based on the reviews, we are likely to accept this manuscript for publication, providing that you modify the manuscript according to the review recommendations. 

The article, entitled “The potential impact of human visceral leishmaniasis (VL) vaccines on population incidence” by the authors Rutte et al brings courtesy to the ongoing VL elimination and the employment of VL transmission model to estimate the potential impact of vaccine characteristics on VL incidence and transmission dynamics during and after the achievement of the current elimination target. Overall, this is a well written manuscript and examines the current progress in vaccine against VL and its impact on VL incidence. There are still minor shortcoming brought out by the reviewers that the authors need to take care in the revised manuscript during their re-submission.

Sincerely,

Angamuthu Selvapandiyan, Ph.D.

Guest Editor

Alvaro Acosta-Serrano

Deputy Editor

The article, entitled “The potential impact of human visceral leishmaniasis (VL) vaccines on population incidence” by the authors Rutte et al brings courtesy to the ongoing VL elimination and the employment of VL transmission model to estimate the potential impact of vaccine characteristics on VL incidence and transmission dynamics during and after the achievement of the current elimination target. Overall, this is a well written manuscript and examines the current progress in vaccine against VL and its impact on VL incidence. There are still minor shortcoming brought out by the reviewers that the authors need to take care in the revised manuscript during their re-submission.

Reviewer's Responses to Questions

**Key Review Criteria Required for Acceptance?**

**Methods**

-Are the objectives of the study clearly articulated with a clear testable hypothesis stated?

-Is the study design appropriate to address the stated objectives?

-Is the population clearly described and appropriate for the hypothesis being tested?

-Is the sample size sufficient to ensure adequate power to address the hypothesis being tested?

-Were correct statistical analysis used to support conclusions?

-Are there concerns about ethical or regulatory requirements being met?

Reviewer #1: yes

Reviewer #2: objectives well defined. Study design not applicatble as it is matehmetical simulation. Popluation cleraly defined to Indian endemic population. Sample size is not applicable. Authors have honestly described limiations in their moddeling assumptions so that helps readers. There are not ethical conserns in the study

Reviewer #3: The objectives of this paper are to explore the utility of vaccines with various characteristics for visceral leishmaniasis through mathematical compartmental model-based simulations. The results are applicable to areas where the disease is anthroponotic. Since this is simulation-based, a number of reasonable (yet restrictive, as noted by the authors) assumptions are made about the hypothetical population and the performance of the vaccine. This is an appropriate approach to study this problem to evaluate the potential effects of a vaccine.

The authors discuss the role that canine vaccines have played in lowering VL incidence in humans and canines. Although not cited, additional canine VL vaccine studies have been conducted in the United States, where the disease is enzootic in hunting hounds but not present in humans (e.g. https://doi.org/10.1016/j.vaccine.2018.08.087). The primary mode of transmission here is vertical, so this would be an interesting addition to the discussion on the importance of vaccines.

**Results**

-Does the analysis presented match the analysis plan?

-Are the results clearly and completely presented?

-Are the figures (Tables, Images) of sufficient quality for clarity?

Reviewer #1: Yes

Reviewer #2: Results are clearly presented as per the typical simulation exercisise.

Reviewer #3: The results for the planned analysis are clearly presented; figures are helpful in communicating results. 

It wasn't obvious to me if the code for running these simulations was available. If it is not, it would be helpful to some readers to have the code available. If it is available, perhaps its location can be made more obvious.

**Conclusions**

-Are the conclusions supported by the data presented?

-Are the limitations of analysis clearly described?

-Do the authors discuss how these data can be helpful to advance our understanding of the topic under study?

-Is public health relevance addressed?

Reviewer #1: Yes

Reviewer #2: Conlusions support data presented as authors do show different implication in out put as per different assumptions and authors do discuss well the implications of their study to public health and and its public health impact.

Reviewer #3: All conclusions are supported by the simulations, and the limitations of the analysis are articulated clearly. The authors communicate the applicability of the simulation results and make a strong case for the utility of a VL vaccine on the Indian subcontinent. 

In the limitations (line 339), the authors state, "...the role of chance increases and a stochastic IBM would be required..." While the need for a stochastic model, particularly in this setting, is apparent, it is not clear why it needs to be an individual based model. It would be helpful to clarify this point.

**Editorial and Data Presentation Modifications?**

Reviewer #1: (No Response)

Reviewer #2: authors can be requested to share their primary data so that other peers interesrted in mathematical moddeling.

Reviewer #3: 1. Pg. 2 Line 28: “… elimination of VL as a public health problem.”

2. Pg. 5 Line 91: “Should an effective vaccine…”

3. Pg. 15 Line 319: “… decided to simulate the vaccine…”

**Summary and General Comments**

Reviewer #1: The article by Rutte et al. entitled “The potential impact of human visceral leishmaniasis vaccines on population incidence” draws attention to the ongoing VL elimination program in India and implementation of VL transmission model to estimate the potential impact of vaccine characteristics on VL incidence and transmission dynamics during and after the achievement of the current elimination target. Overall, this manuscript is well written and discusses the current progress in VL vaccine and its impact on VL incidence. However, there are some issues that need to be revised in this current version of the manuscript, to justify the publication in this journal, as follows:

1) I am in complete agreement with the author's interpretation of the challenges and actions needed for sustainable VL elimination. They rightly point out the huge gap in knowledge concerning transmission and the reservoir and without this information it will be very difficult to go the last mile and reach sustainable elimination. Importantly, I am not sure about availability of any human vaccine in coming near future. Therefore, it would be interesting if mathematical modelling people provide some information about expected deadline of elimination using current tools (without vaccine).

2) The elimination of VL in South Asia has been qualified as elimination to a level where it is not a public health problem. Therefore this definition of elimination is quite different to the classical definition of elimination where there is no local transmission. This point should be made clear in the text as the approach after the elimination would be quite different in the 2 situations. In several places it seems they have not been able to keep this in mind.

3) Authors have clearly mentioned in the manuscript that contribution of asymptomatics to disease transmission has yet not been identified (line 68-69), however, they consider asymptomatic subjects as major contributor to transmission in the model variant (Table-2). It is therefore important here to justify the data with evidence.

4) Figure-1 is missing HIV-VL co infection subjects as reservoirs.

Reviewer #2: Some of the assumptions are too simplisting however authors can not be faulted as there biology and infectious epidemiology of VL is still have lots of gaps in current understanding. Overall as a reviewver i feels that authros have done good job in carryin out this work inspite of limited and scanty knowledge available.

Reviewer #3: (No Response)

PLOS authors have the option to publish the peer review history of their article (what does this mean?). If published, this will include your full peer review and any attached files.

Reviewer #1: No

Reviewer #2: Yes: Rajan R Patil

Reviewer #3: No
---

## [Editor Report · Decision Letter 1]

10 Jun 2020

Dear Dr Le Rutte,

We are pleased to inform you that your manuscript 'The potential impact of human visceral leishmaniasis vaccines on population incidence' has been provisionally accepted for publication in PLOS Neglected Tropical Diseases.

Best regards,

Angamuthu Selvapandiyan, Ph.D.

Guest Editor

Alvaro Acosta-Serrano

Deputy Editor

The authors have addressed very well to the comments/suggestions raised by the reviewers and edited the manuscript accordingly.

---

## [Editor Report · Acceptance letter]

24 Jun 2020

Dear Dr Le Rutte,

We are delighted to inform you that your manuscript, "The potential impact of human visceral leishmaniasis vaccines on population incidence," has been formally accepted for publication in PLOS Neglected Tropical Diseases.

Best regards,

Shaden Kamhawi

co-Editor-in-Chief

Paul Brindley

co-Editor-in-Chief
